# Investigation of mobile genetic elements and their association with antibiotic resistance genes in clinical pathogens worldwide

Markus H. K. Johansson[1]*, Thomas N. Petersen[1], Sidsel Nag[1],
Timmie M. R. Lagermann[1], Laura E.K. Birkedahl[1], Silva Tafaj[2], Susan Bradbury[3],
Peter Collignon[3], Denise Daley[4], Victorien Dougnon[5], Kafayath Fabiyi[5],
Boubacar Coulibaly[6], Réné Dembélé[7], Natama Magloire[8], Isidore J. Ouindgueta[9],
Zenat Z. Hossain[10], Anowara Begoum[10], Deyan Donchev[11], Mathew Diggle[12],
LeeAnn Turnbull[12], Simon Lévesque[13], Livia Berlinger[14], Kirstine K. Søgaard[15],
Paula D. Guevara[16], Carolina Duarte[16], Panagiota Maikanti[17], Jana Amlerova[18],
Pavel Drevinek[19], Jan Tkadlec[19], Milica Dilas[20], Achim Kaasch[20], Henrik T. Westh[21],
Mohamed A. Bachtarzi[22], Wahiba Amhis[22], Carolina E.S. Salazar[23], José E. Villacis[24],
Mária A. D. Lúzon[25], Dàmaris B. Palau[26], Claire Duployez[27], Maxime Paluche[28],
Solomon Asante-Sefa[29], Mie Møller[30], Margaret Ip[31], Ivana Mareković[32],
Agnes Pál-Sonnevend[33], Clementiza E. Cocuzza[34], Asta Dambrauskiene[35],
Alexandre Macanze[36], Anelsio Cossa[36], Inácio Mandomando[36],
Philip Nwajiobi-Princewill[37], Iruka N. Okeke[38], Aderemi O. Kehinde[39,40], Ini Adebiyi[38,40],
Ifeoluwa Akintayo[39], Oluwafemi Popoola[39,40], Anthony Onipede[41], Anita Blomfeldt[42],
Nora E. Nyquist[42], Kiri Bocker[43], James Ussher[43], Amjad Ali[44], Nimat Ullah[44],
Habibullah Khan[45], Natalie W. Gustafson[46], Ikhlas Jarrar[47], Arif Al-Hamad[48],
Viravarn Luvira[49], Wantana Paveenkittiporn[50], Irmak Baran[51], James C. L. Mwansa[52],
Linda Sikakwa[53], Kaunda Yamba[54], Frank M. Aarestrup[1]

1 National Food Institute, Technical University of Denmark, Kgs. Lyngby, Denmark, 2 Microbiology Department, University Hospital "Shefqet Ndroqi", Tirana, Albania, 3 Microbiology Department, Canberra Hospital, Garran, Australian Capital Territory, Australia, 4 Department of Microbiology, PathWest Laboratory Medicine, Fiona Stanley Hospital, Murdoch, Western Australia, Australia, 5 Research Unit in Applied Microbiology and Pharmacology of Natural Substances, Polytechnic School of Abomey-Calavi, University of Abomey-Calavi, Abomey-Calavi, Cotonou, Benin, 6 Department of Laboratory, Nouna Health Research Centre, Nouna, Burkina Faso, 7 Training and Research Unit in Applied Sciences and Technologies/Biochemistry-microbiology, University of Dedougou, Dedougou, Boucle du Mouhon, Burkina Faso, 8 Clinical Research Unit of Nanoro, National Institutes of Medical Research, Ouagadougou, Burkina Faso, 9 University of Joseph KI-ZERBO, Ouagadougou, Burkina Faso, 10 Department of Microbiology, University of Dhaka, Dhaka, Bangladesh, 11 Clinical Laboratory of Microbiology and Virology, University Hospital "Lozenetz", Sofia, Bulgaria, 12 Alberta Precision Laboratories, Alberta, Canada, 13 Service de Microbiologie, Centre Integré Universitaire de Santé et de services sociaux de l'Estrie – Centre Hospitalier Universitaire de Sherbrooke, Sherbrooke, Québec, Canada, 14 Bioanalytica AG, Luzern, Switzerland, 15 Division of Clinical Bacteriology and Mycology, University Hospital Basel, Basel, Switzerland, 16 Microbiology Group, Instituto Nacional de Salud, Bogotá, Colombia, 17 Charalampous, Microbiology Department, National Reference Laboratory for Antimicrobial Resistance Surveillance, Nicosia General Hospital, Strovolos, Nicosia, Cyprus, 18 Department of Microbiology, University Hospital in Plzen, Plzen, Czech Republic, 19 Department of Medical Microbiology, Motol University Hospital, Prague, Czech Republic, 20 Otto-von-Guericke University, Magdebourg, Germany, 21 Klinisk Mikrobiologisk Afdeling, Hvidovre Hospital, Hvidovre, Denmark, 22 Laboratoire de Microbiologie Clinique, Centre Hospitalo-universitaire, Algiers, Algeria, 23 National Reference Center for Antimicrobial Resistance, National Institute of Public Health research "Dr. Leopoldo Izquieta Pérez", Quito, Pichicha, Ecuador, 24 Centro de Investigación para la Salud en América Latina (CISeAL), Pontificia Universidad Católica del Ecuador, Quinto, Pichincha, Ecuador, 25 Department of Pathology and Experimental Therapy, Universitat de Barcelona, Barcelona, Spain, 26 Microbiology Department, Hospital de Bellvitge, Barcelona, Spain, 27 Institute of Microbiology, Centre Hospitalier Universitaire de Lille, Lille, France, 28 Bacteriology laboratory, Centre hospitalier de Valenciennes, Valenciennes, France, 29 Sekondi Public Health Laboratory, Ghana Health Service, Effia Nkwanta Regional Hospital, Effia Nkwanta Regional Hospital,



**Data availability statement:** All sequenced genomes are available from the European Nucleotide Archive database (https://www.ebi.ac.uk/ena/browser/view/ERP141886).

**Funding:** This work was supported by the Novo Nordisk Foundation (Grant: NNF16OC0021856: Global Surveillance of Antimicrobial Resistance) and the European Union's Horizon 2020 research and innovation program (Grant: 874735). The funders had no role in study design, data collection and analysis, decision to publish, or preparation of the manuscript.

**Competing interests:** The authors have declared that no competing interests exist.

Ghana, **30** Dronning Ingrids Hospital, Nuuk, Greenland, **31** Chinese University of Hong Kong, Shatin, Hong Kong, **32** Department of Clinical and Molecular Microbiology, University Hospital Centre Zagreb, Zagreb, Croatia, **33** Medical Microbiology and Immunology, University of Pecs Medical School, Pecs, Hungary, **34** Department of Medicine and Surgery, University of Milano-Bicocca, Milan, Italy, **35** Laboratory Medicine Department, Hospital of Lithuanian University of Health Sciences, Kaunas, Lithuania, **36** Centro de Investigação em Saéude de Manhiça, Manhiça, Mozambique, **37** National Hospital Abuja, Abuja, Nigeria, **38** Faculty of Pharmacy, University of Ibadan, Ibadan, Oyo State, Nigeria, **39** College of Medicine, University of Ibadan, Ibadan, Oyo State, Nigeria, **40** University College of Ibadan, Ibadan, Oyo State, Nigeria, **41** Obafemi Awolowo University, Ile-Ife, Nigeria, **42** Department of Microbiology and Infection Control, Akershus University Hospital, Lørenskog, Norway, **43** Southern Community Laboratories, University of Otago, Otago, Dunedin, New Zealand, **44** Department of Industrial Biotechnology, Atta-ur-Rahman School of Applied Biosciences, National University of Sciences and Technology (NUST), Islamabad, Pakistan, **45** Molecular Diagnostic Section, Khyber Teaching Hospital (KTH), Peshawar, Pakistan, **46** Departamento de Bacteriologia, Laboratorio Central de Salud Publico, Asunción, Paraguay, **47** Basic Medical Sciences Department, Arab American University, Jenin, Palestine, **48** Division of Clinical Microbiology, Qatif Central Hospital, Al-Qatif, Eastern Province, Saudi Arabia, **49** Department of Clinical Tropical Medicine, Faculty of Tropical Medicine, Mahidol University, Bangkok, Thailand, **50** Department of Medical Sciences, National Institute of Health, Sariburi, Thailand, **51** Medical Microbiology Department, Karadeniz Technical University Farabi Hospital, Trabzon, Ortahisar, Turkey, **52** Lusaka Apex Medical School, Lusaka, Zambia, **53** Levy Mwanawasa Teaching Hospital, Lusaka, Zambia, **54** University Teaching Hospital, Lusaka, Zambia

\* markjo@food.dtu.dk

## Abstract

### Objectives

Antimicrobial-resistant bacteria are a major global health threat. Mobile genetic elements (MGEs) have been crucial for spreading resistance to new bacterial species, including human pathogens. Understanding how MGEs promote resistance could be essential for prevention. Here we present an investigation of MGEs and their association with resistance genes in pathogenic bacteria collected from 59 diagnostic units during 2020, representing a snapshot of clinical infections from 35 counties worldwide.

### Methods

We analysed 3,095 whole-genome sequenced clinical bacterial isolates from over 100 species to study the relationship between resistance genes and MGEs. The mobiliome of *Staphylococcus aureus*, *Enterococcus faecalis*, *Escherichia coli*, and *Klebsiella pneumoniae* were further examined for geographic differences, as these species were prevalent in all countries. Genes potentially mobilized by MGEs were identified by finding DNA segments containing MGEs and ARGs preserved in multiple species. Network analysis was used to investigate potential MGE interactions, host range, and transmission pathways.

### Results

The prevalence and diversity of MGEs and resistance genes varied among species, with *E. coli* and *S. aureus* carrying more diverse elements. MGE composition differed between bacterial lineages, indicating strong vertical inheritance. 102 MGEs associated with resistance were found in multiple species, and four of these elements seemed to be highly transmissible as they were found in different phyla. We identified 21 genomic

regions containing resistance genes potentially mobilized by MGEs, highlighting their importance in transmitting genes to clinically significant bacteria.

## Conclusion

Resistance genes are spread through various MGEs, including plasmids and transposons. Our findings suggest that multiple factors influence MGE prevalence and their transposability, thereby shaping the MGE population and transmission pathways. Some MGEs have a wider host range, which could make them more important for mobilizing genes. We also identified 103 resistance genes potentially mobilised by MGEs, which could increase their transmissibility to unrelated bacteria.

## Introduction

The emergence of antimicrobial-resistant bacteria (AMR) is recognised as a significant threat to global public health [1,2] and estimates suggest that 1.3 million deaths annually can be attributed to AMR [3].

Several bacteria, such as the ESKAPE [4] group pathogens (*Enterococcus faecium, Staphylococcus aureus, Klebsiella pneumoniae, Acinetobacter baumanii, Pseudomonas aeruginosa,* and *Enterobacter* spp) and *Escherichia coli* have been increasingly involved in resistant infections [5]. These bacteria can be difficult to treat since they frequently carry antimicrobial resistance genes (ARG), often acquired via horizontal gene transfer and mediated by various mobile genetic elements (MGE) [6–9].

MGEs are discrete regions of DNA that can promote their transposition or the transposition of other elements between DNA molecules. They are classified based on their properties and genetic layout into types [6]. Inter-cellular transposing MGEs such as plasmids, integrative and conjugative elements (ICE), and integrative and mobilisable elements (IME) can either conjugate or be mobilised by the conjugation of other elements [7–9]. They often carry other ARGs or intra-cellular transposing MGEs. Unlike plasmids, ICEs and IMEs are primarily integrated into the host chromosome [10]. MGEs integrated into a host's DNA are referred to as integrated MGEs (iMGEs).

Insertion sequences (IS) are among the smallest types of intra-cellular transposing iMGEs, often consisting of a transposase bounded by inverted repeats (IRs) [11,12]. They are typically unable to carry accessory genes but can facilitate gene mobility by forming composite transposons (comTn) structures [13, 14]. Unit transposons (Tn) are a diverse type of iMGEs whose members are frequently associated with ARGs. Several members carry integrons [15,16], a type of iMGE that can rapidly exchange their carried accessory genes [17]. Extensive literature describes the different types of MGEs [6,9] and their association with AMR in various species [18–20].

The interplay of different MGEs forms a complex transposition network that has been essential for recruiting ARGs [7, 21] and spreading them to infectious bacteria [22,23]. MGEs are also thought to help retain resistance genes in environments with low levels of antimicrobials [24].

Here, we present an investigation of resistance determinants and their association with MGE from 59 diagnostic units worldwide. Samples were collected in 2020 from patients with symptoms of disease caused by a pathogen, and thus, whole genome sequencing was used for diagnostic analysis. These samples provide a valuable insight into the association between ARGs and MGEs as they represent an unbiased snapshot of clinically relevant bacteria isolated in 2020. The data encompasses a diverse set of bacterial species, lineages, and tissue types and should therefore better approximate the diversity of resistance determinants and mobilome.

## Methods

### Dataset selection

Clinical pathogens were collected as part of the Two Weeks in the World (TWIW) research project by participating diagnostic units during 2020 and sent to the Group for Genomic Epidemiology, Technical University of Denmark (DTU), for

whole genome sequencing. Protocols for DNA sampling are available at the TWIW website (https://twiw.genomicepide-miology.org/about), and the sequences are available at ENA (see S1 Appendix for sample information and accession numbers). The TWIW dataset comprises 3,095 sequenced isolates, representing 115 species from 35 countries [25].

We primarily investigated *E. coli*, *K. pneumoniae*, *E. faecium*, and *S. aureus* isolates as they were the most prevalent Gram- and Gram+ species in this dataset with a good geographic representation. The sequences were processed, quality-controlled, and *de-novo* assembled using the method outlined in the supplementary methods section. A total of 1,191 samples (553 *E. coli*, 227 *K. pneumoniae*, 77 *E. faecium*, 334 *S. aureus*), representing 35 countries from six WHO-defined regions [23] and is now referred to as the "primary" dataset.

The remaining sequences from less frequent species were used to verify putative transposable elements and to investigate the phylogenetic confinement of iMGEs. These sequences were processed using the previously described method, creating a "secondary dataset" consisting of 1,559 samples from 101 species.

## Read processing, genome assembly, and QC filtering

Only isolates sequenced using the Illumina NextSeq 500 platform were included to mitigate bias introduced by differences in sequencing chemistry.

Reads were trimmed using bbduk2 (part of BBmap version 36.49) (score cutoff 20 and removing reads shorter than 50 bp), and adapter sequences were removed by matching using an internal adapter database [26]. Read quality was assessed with FastQC [27] (version 0.11.5) before and after read processing. The processed reads were *de-novo* assembled with Spades [28] version 3.11.0 using error correction, coverage cutoff = 2, and k-mer sizes 21, 33, 55, 77, 99, and 127. Assembly quality metrics were calculated with Quast [29] version 4.5. If the isolate had been sequenced multiple times (47 isolates), the sequencing run with the highest-quality bases was selected.

Kmerfinder [30] and ribosomal Multi Locus Sequence Typing (rMLST) were used to verify project participants' species annotations and identify contaminated samples. Isolates in which the predicted species differed from the annotation were excluded, as they could have been mixed up.

Kmerfinder and rMLST were run using the default thresholds using the processed reads as input. A species was said to be verified if Kmerfinder and rMLST predictions were in concordance, and the main species prediction exceeded 80% rMLST support and 40% Kmerfinder coverage. Isolates predicted to contain multiple species (Kmerfinder coverage > 1% or rMLST support > 1%) were excluded since they could be contaminated.

Isolates with an unexpectedly large assembly size were removed, as it could indicate that they contained multiple strains, i.e., if the assembly size was larger than 99.7% (3 std) of the reference genomes for that species in NCBI RefSeq. Genome sizes of non-atypical genomes with the completeness of "chromosome" or "complete assembly" were used (data accessed 2023-03-17).

The assembly completeness was estimated from the proportion of non-identified loci in the core genome Multi Locus Sequence Typing (cgMLST) profiles. chewBBACA [31] version 3.0.0 with the relevant schemas from cgmlst.org [32–34] was used for typing. Isolates with < 95% of the core genome were excluded.

A maximum of 15 isolates per species and city was included to reduce the overrepresentation of some locations. The final dataset consisted of 1,191 isolates from 35 countries covering seven geographical world regions.

## Controlling for high relatedness in the dataset

To evaluate the diversity of the dataset and ensure it reflects the strains typically encountered in healthcare settins, we employed a combination of cgMLST and Multi Locus Sequence Typing (MLST) typing. The within-species diversity was estimated using both the cgMLST and MLST profiles. Overrepresented lineages were identified from the MLST sequence types (ST) using the mlst [35,36] tool version 2.23.0. Isolates were clustered based on their cgMLST profile with scipy [23] version 1.10.1 using Jaccard distance and visualised using the toolkit for Python.

Each species was represented by isolates from several MLST sequence types (*E. coli* 94, *K. pneumoniae 95, E. faecalis* 31, and *S. aureus* 56), of which approximately 44%−76% were considered rare as they were found in less than 5% of all isolates (S1 Fig). The high average pairwise cgMLST allele difference for all species also suggested that the dataset is highly diverse.

### *In-silico* prediction of AMRs and iMGEs

ARGs were predicted with Resfinder (tool version 4.3.1, ResfinderDb version 2.1.0, disinfinderDb version 2.0.0) with default settings, using the assembled contigs as input [37]. An updated version of MobileElementFinder [38] (version 1.1.2) was used to predict miniature inverted-repeats transposable elements (MITEs), IS, Tn, ICE, IME, and cis-mobilizable elements (CIMEs). This new version adds 1,686 IS and 70 Tn to its database.

### Identification of plasmid-borne iMGEs and ARGs

Contigs were classified as being of either plasmid or chromosomal origin using the consensus prediction of PlasClass [39] version 0.1.1 (threshold 0.5) and Platon [40] version 1.6 (executed in *accuracy* mode with either the *Enterococcus*, *Staphylococcus*, or Bacteria database).

A two-way ANOVA (statsmodels python package) was used to evaluate differences in the amount of plasmid DNA between the species. The normality of residuals and heterogeneity of variance were ensured.

### Data visualisation

Unless otherwise stated, all visualisations were created using a combination of Matplotlib [41] version 3.7.1 and Seaborn [42] version 0.12.2 for Python 3.11. Geographical maps were created using GeoPandas version 0.13.0 using shapefiles from Natural Earth.

### Investigation of the iMGE population, geography, and lineage

Isolates were clustered using scipy [23] on the iMGE profile (Jaccard distance and average linkage) to investigate differences between geographical regions and lineages.

### Identification of iMGEs associated with ARGs and putative mobilised complexes

iMGEs have the potential to mobilize nearby genes by forming comTns. Potentially mobilized ARGs were identified in the assembled 1,191 primary samples by locating iMGEs within 10 kb of an ARG, which corresponds with the largest compTn in the MGEdb database. Combinations of iMGE and ARG were kept if the distance between the elements and the intermediary sequence was preserved in multiple samples. The sequence similarity was calculated using Sourmash [43] version 4.8.2.

Finding the same preserved combination of iMGE and ARG, or putative translocatable unit (pTU), in multiple species could indicate mobilization. The presence of the pTUs on other species was investigated by searching for similar sequences in the secondary dataset using blastn [44] version 2.14.0. The pTU was queried against a custom database of the assembled genomes, and hits with greater than 95% query coverage and greater than 90% sequence identity were considered matches.

### Investigation of the iMGE host range

The presence of iMGEs in bacterial species was used to estimate host range and possible transposition pathways. iMGEs were said to be confined to a taxonomic level if they were only found in bacteria of the same taxa. Hypothetical transposition pathways were modelled as a bi-directional multi-graph where species constituted nodes and iMGEs edges using NetworkX [45] version 3.1. Communities of densely connected bacterial species were identified using the Louvain method [46] of community detection.

## Results

Here, we present an analysis of mobilised ARGs in 1,191 clinical isolates (primary dataset) from the four common pathogens *E. coli* (553), *K. pneumoniae* (227), *E. faecalis* (77), and *S. aureus* (334), representing a snapshot of the global diversity of resistance determinants (S1 Appendix).

The data is a subset of a larger dataset containing 3,095 clinical isolates, including 115 species worldwide [47] as seen in S2 Fig. All isolates are available on ENA (accession number: ERP141886).

### Species-specific differences in the ARG profile

A total of 7,159 ARGs were identified in the primary dataset. Among the four pathogens, Gram- bacteria exhibited greater ARG diversity, with *K. pneumoniae* carrying the most diverse set of resistance genes (S3b Fig). Beta-lactam resistance genes were highly prevalent in *K. pneumoniae* (99.6% of all isolates), followed by *S. aureus* (79.3%) and *E. coli* (58.6%). In contrast, no beta-lactam resistance genes were in any of the *E. faecalis* isolates, and in *S. aureus,* it was primarily mediated by *bla*Z and *mec*A.

The distribution of beta-lactamase genes varied geographically, particularly in *E. coli* and *K. pneumoniae.* In *E. coli*, $bla_{TEM}$ genes were widespread across all continents, whereas $bla_{SHV}$ was more prevalent in central Europe. The $bla_{CTX-M}$ and $bla_{OXA}$ gene families were more prevalent in southern Europe, Turkey, and Africa (Fig 1).

Geographic differences were also evident in the distribution of gene variants. For instance, $bla_{CTX-M-15}$ and $bla_{CTX-M-27}$ were found on almost every continent, whereas $bla_{CTX-M-1}$ was only found in Europe, and $bla_{CTX-M-24}$ was only identified in isolates from Oceania (S1 Table). The $bla_{CMY}$ genes were only found in isolates from Asian and African countries, with Asian isolates exhibiting higher gene diversity than those observed in Africa. In contrast, African isolates only carried $bla_{CMY-2}$.

In *K. pneumoniae*, beta-lactam resistance was primarily mediated by $bla_{SHV}$, $bla_{TEM}$, and $bla_{CTX}$ genes. Notably, isolates from South America frequently harboured primarily $bla_{KPC}$ and $bla_{NDM}$ genes (Fig 2 and S2 Table), highlighting regional differences in resistance mechanisms.

Approximately 75% of the *E. faecalis* isolates were predicted to carry tetracycline resistance genes (mediated by *tet*(M) and *tet*(L)), while about 44.1% of the isolates carried aminoglycoside resistance genes. Macrolide (~35.1%), glycopeptides (~2.6%), and amphenicol (~1.2%) resistance genes were only present in a small number of isolates (S4 Fig). The intrinsic *Lsa* family of efflux pumps [48] was found in all isolates.

ARGs varied between the different species, just like with iMGEs. For example, $bla_{TEM-1B}$ and *dfr*A17 were more prevalent in *E. coli*, while *fos*A, *Oqx*A, and *Oqx*B were more common in *K. pneumoniae*. There were only minor differences in the ARG profile between MLST ST. *E. faecalis* was the exception where st480 harboured macrolide resistance gene *dfr*G and lincosamide resistance gene *lnu*(B) compared to st6 and st774, which tended to carry the aminoglycoside resistance gene ant(6)-Ia, as can be seen in S5 Fig. Only minor differences in ARG prevalence between continents, as seen in S6 Fig.

### Identification of transferable ARGs

The contigs were classified as being of either plasmid or chromosomal origin based on the consensus prediction of PlasClass and Platon. *S. aureus* had significantly lower estimated plasmid content than the other species (P-value $2.36 \times 10^{-129}$; statistics in S3 Table), averaging 15.9 Kbp plasmid DNA per genome (~0.6% of the assembled DNA). *E. coli, K. pneumoniae,* and *E. faecalis* had comparable plasmid content relative to the total assembled DNA (~2.3%,~3.8%, and ~2.7%), S7 Fig.

ARGs mobilised by plasmids were identified as they could spread to other bacteria. The number of mobilised ARGs varied between the species, with plasmid-borne ARGs being more common in Gram- bacteria than in Gram+ ones (Fig 3). Most ARGs in *E. faecalis* were located on the chromosome, except for Amphenicol and Glycopeptide resistance genes. *S. aureus* carried most Lincosamide, Streptogramin b, and tetracycline resistance genes on plasmids (S8 Fig). The

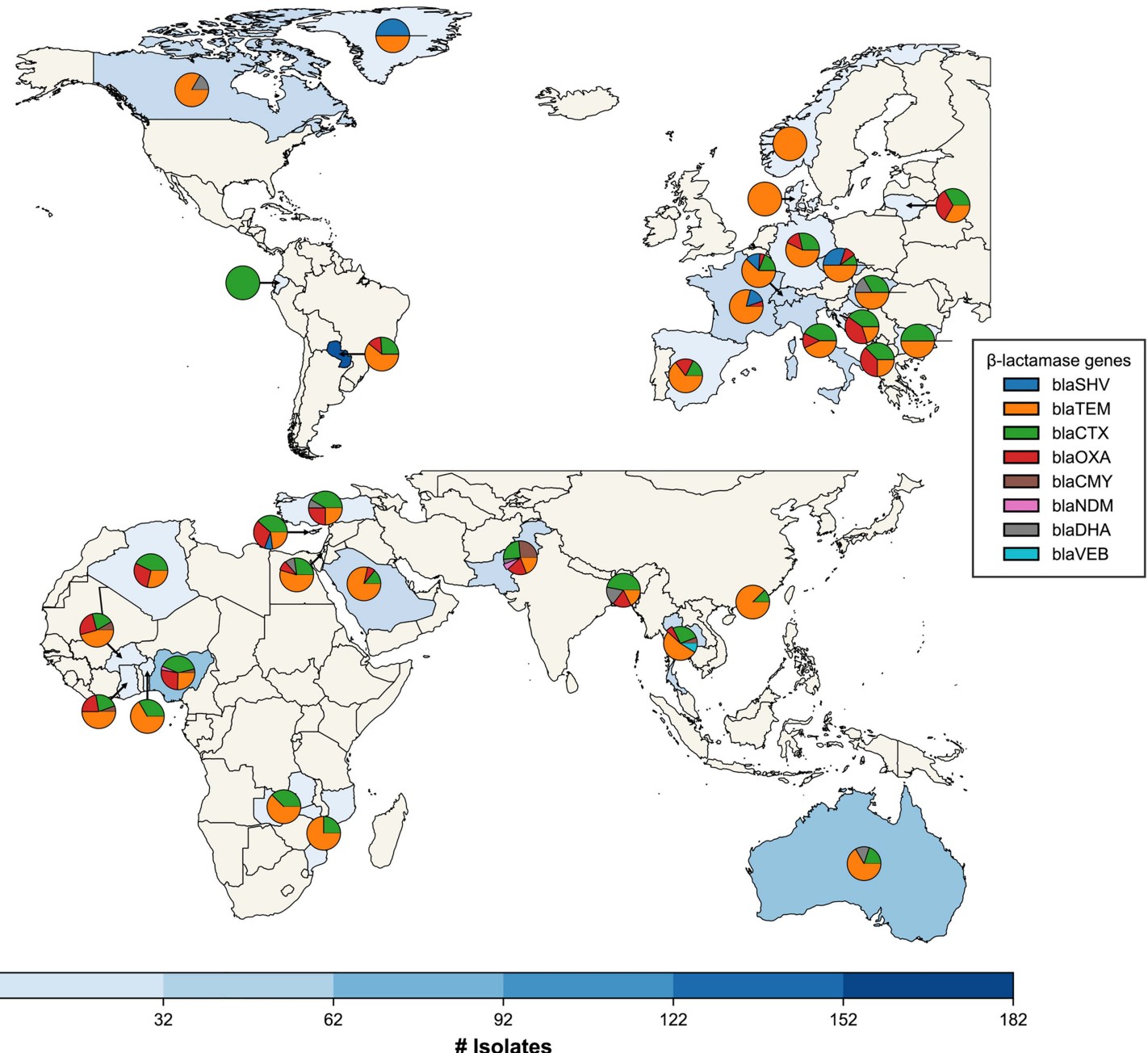

**Fig 1. Geographic distribution of beta-lactamase gene families in *E. coli*.** Countries with resistance are coloured in blue. The pie chart shows the composition of gene families per country. Map was created using shapefiles from Natural Earth.

Gram- species carried most ARGs on plasmids regardless of AMR class (Fig 3) except for the $bla_{SHV}$ family of beta-lactamases in *K. pneumoniae* that were primarily located on chromosomal contigs (S2 Appendix).

ARGs mobilised by iMGEs were also identified as they could facilitate the spread of resistance. Of the 7,159 predicted ARGs, 327 were carried by iMGEs of different types. *S. aureus* and *E. coli* had the highest abundance of iMGEs mobilised ARGs (Fig 4).

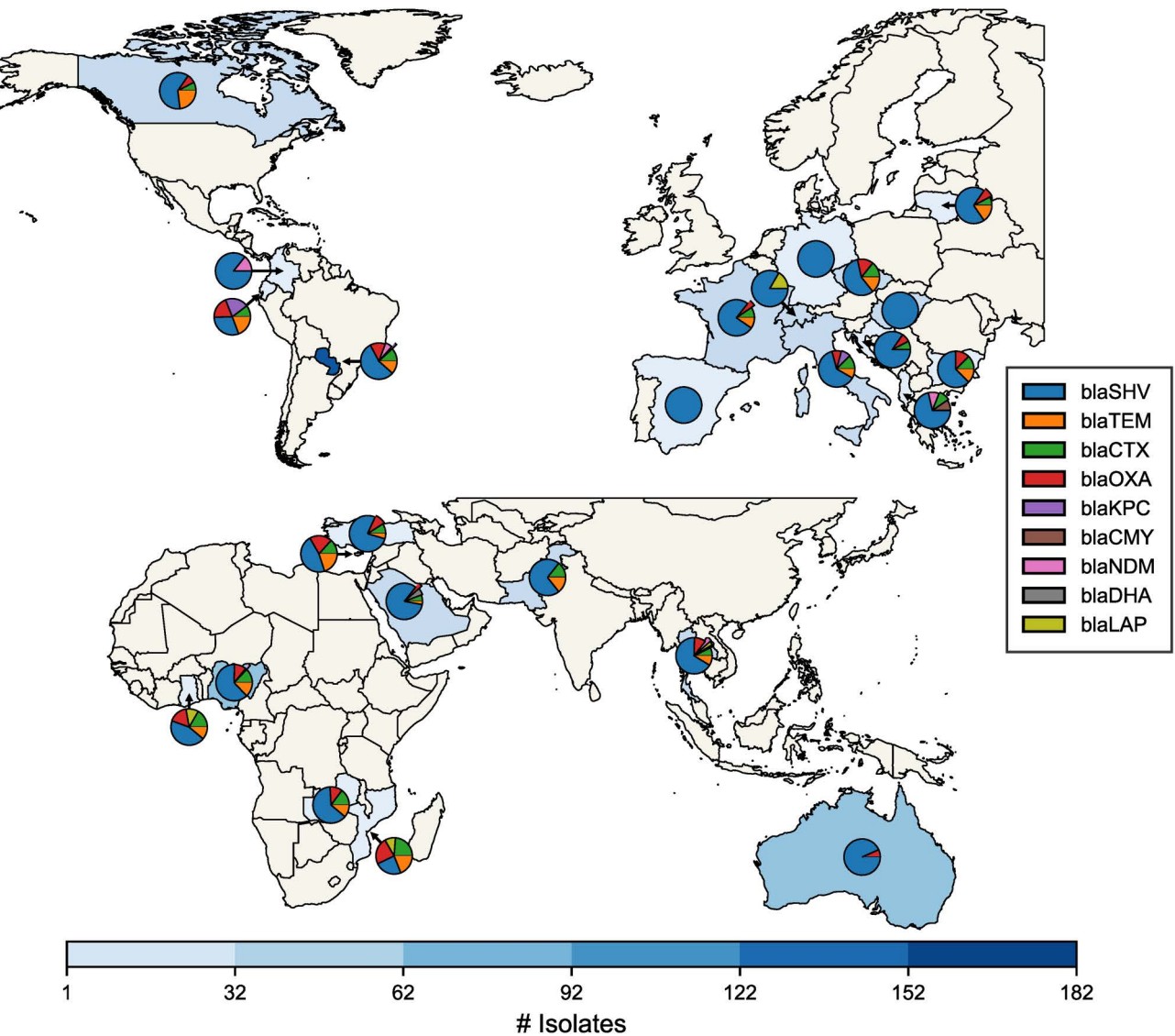

**Fig 2. Geographic distribution of beta-lactamase gene families in *K. pneumonia*.** Countries with resistance are coloured in blue. The pie chart shows the composition of gene families per country. Map was created using shapefiles from Natural Earth.

Beta-lactamases were commonly associated with iMGEs in all species except *E. faecalis*. In *S. aureus blaZ* was mainly carried by the Tn*552* unit-transposon, but in six cases it was found interspersed between copies of the IS ISSep*1* or ISSau*6*. These MGEs could potentially form a comTn.

Various $bla_{TEM}$ beta-lactamases were carried by Tn*2* and Tn*801* unit-transposons in *E. coli* and *K. pneumoniae.* Other beta-lactamase genes, such as $bla_{KPC-2}$, $bla_{CTX-M-15}$, and $bla_{SHV-1,}$ were instead located between copies of ISEc*9*, ISEc*26*, and ISKpn*31* that potentially could transpose the genes by forming a comTn. Despite *E. faecalis* harbouring a larger diversity and higher relative abundance of iMGEs (Fig 4) than the other species, only three iMGE were identified to mobilise ARGs. These ARGs and iMGEs were: *erm*(B) carried by Tn*917, lnu*(G) carried by Tn*6260*, and the Aminoglycoside resistance gene aac(6')-aph(2'') harboured by an IS*256*-based comTn.

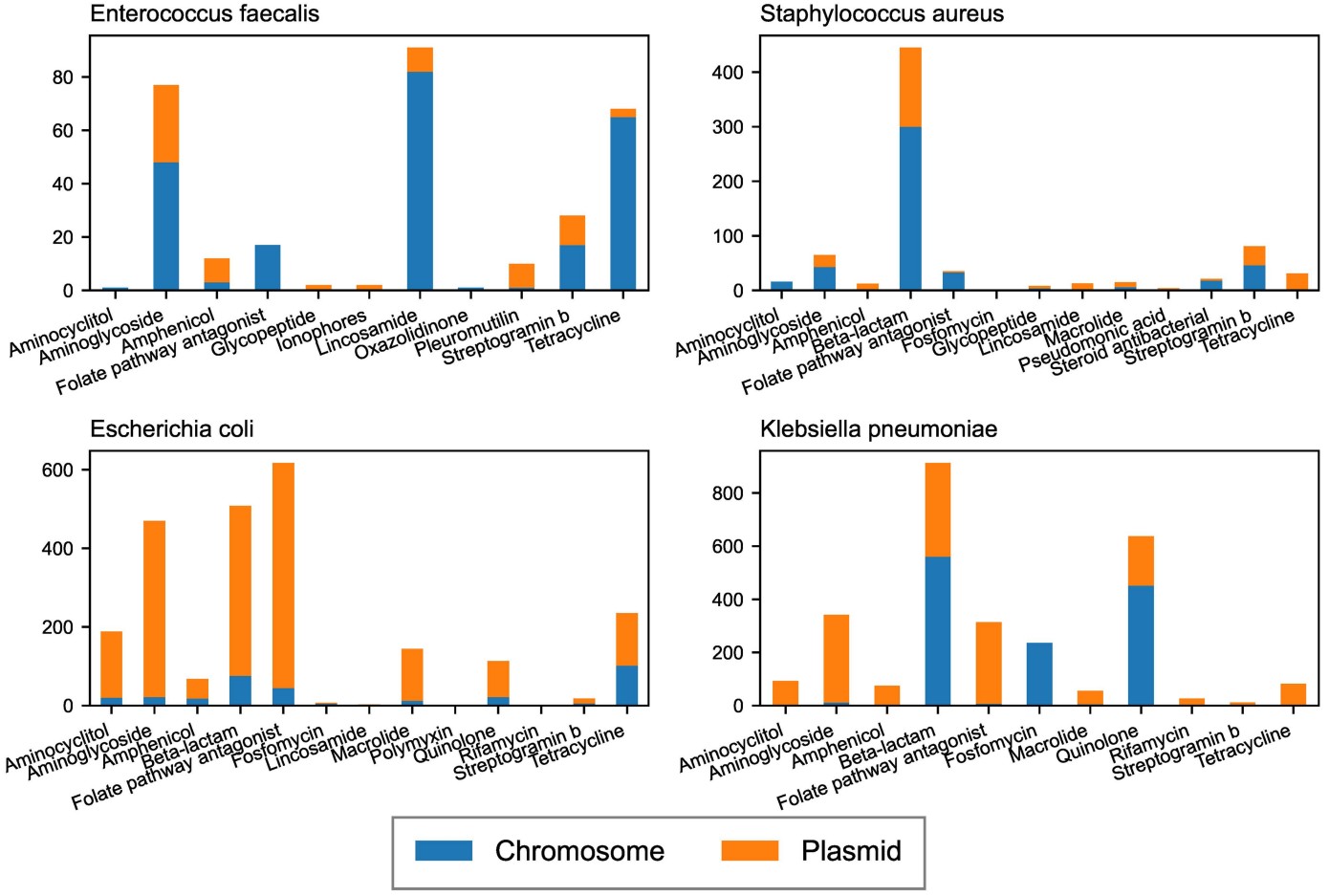

**Fig 3. Number of ARGs per species and their predicted location.**

## iMGEs associated ARGs indicate cross-species transposition

We found evidence of 21 MGEs and ARGs being transposed as a unit because they were repeatedly found at the same distance in multiple species in either the primary or secondary dataset. Eight of these pTUs were found on both plasmid and chromosomal contigs in two of the species, suggesting intra-cellular mobility. The secondary dataset was searched to confirm whether the pTUs were present in additional species beyond the four species in the primary dataset. Most pTUs were found in additional genera, further supporting the hypothesis that they have been transposed as a conserved unit. For example, ISEc9-bla$_{CTX-M-15}$ was found in multiple genera, including *Enterobacter*, *Escherichia*, *Klebsiella*, and *Serratia* (Fig 5). This indicates the ability of MGEs to transmit genes between bacteria of clinical importance.

## Species-specific differences in the MGE population

A total of 19,752 iMGEs were identified in all four species. Gram- bacteria, *E. coli* and *K. pneumoniae*, were found to harbour a larger abundance and diversity of known iMGEs (Fig 1 and S3 Fig) compared to the Gram+ *S. aureus* and *E. faecalis*. IS was the most common type of iMGEs in all species, but the proportion of plasmid-borne elements was higher in Gram- bacteria (Fig 6). MITEs and IME types of iMGEs were only found in *E. coli* and *K. pneumoniae*.

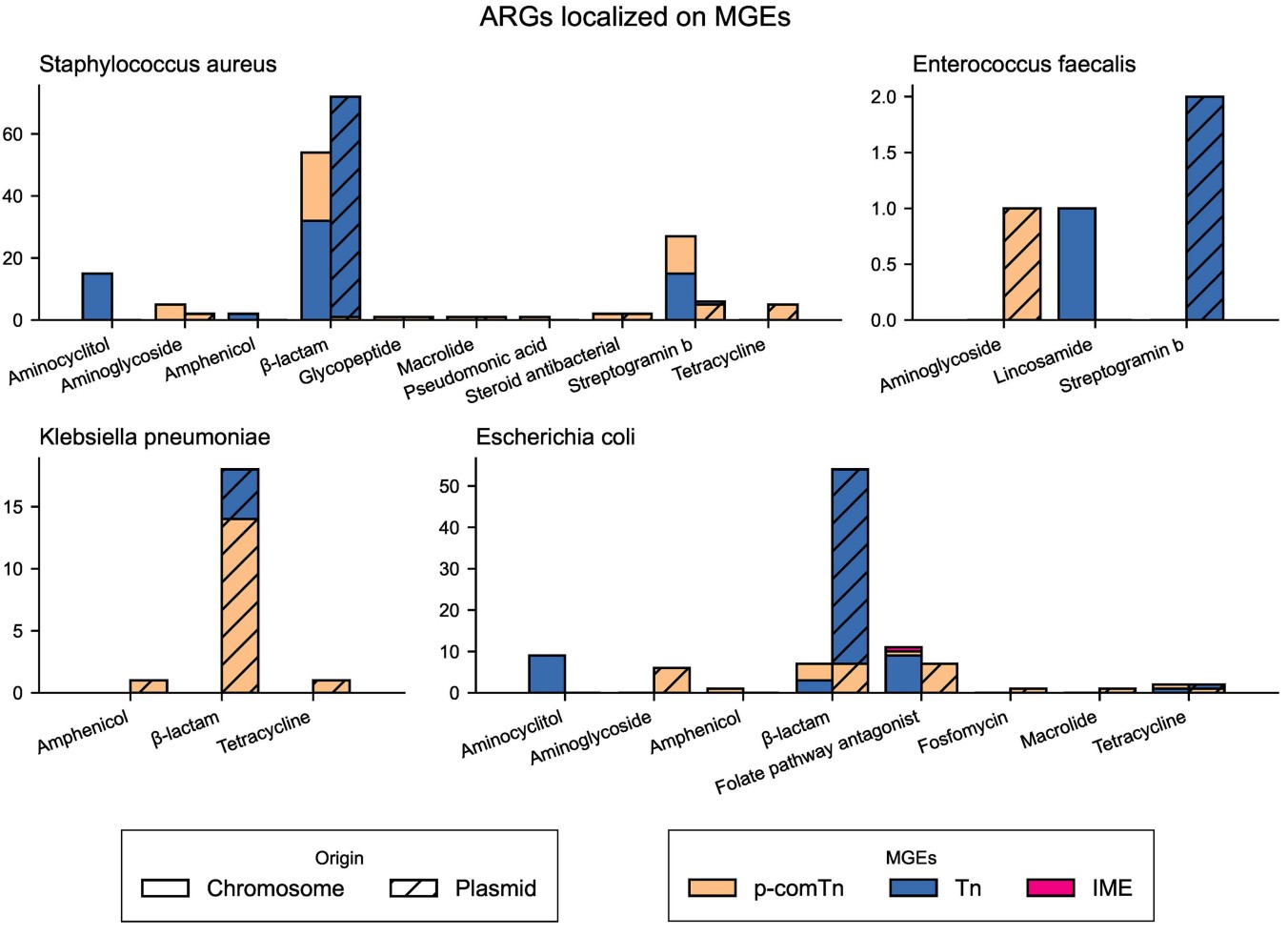

**Fig 4. Number of ARGs mobilized by iMGEs per species.** Hatched bars denote that the ARG is located on a plasmid.

There was a systematic difference in the iMGEs population where most elements were only found in the same species, as can be seen in S9 Fig. For example, insertion sequences ISEc1, IS3, and IS609 were exclusively found in *E. coli,* while ISKpn38, ISKox1, and ISEc15 were only predicted in *K. pneumoniae* (S3 Appendix). Similarly, ISAu6 and ISSep3 and the transposon Tn554 were only found in *S. aureus,* while the insertion sequences ISLmo19, IS1062, and Tn6260 were only identified in *E. faecalis* (S3 Appendix).

### iMGE profile corresponds with lineage

Isolates of the same ST tended to carry similar iMGEs. The relationship between iMGE profile and lineage was most pronounced in *E. coli,* where several STs (st3, st4, and st53) corresponded with isolated iMGE clusters. The relationship was also strong for *S. aureus*, *K. pneumoniae*, and *E. faecalis*. The clusters are presented in S10-S14 Figs.

Some lineages tended to share similar iMGE profiles. For instance, st506 and st43 in *E. coli,* st258 and st11 in *K. pneumoniae,* and st5 and st45 in *S. aureus* are samples with similar iMGE profiles. Some STs (st1 in *E. coli* and st11 in *K. pneumoniae*) consisted of multiple iMGE clusters, indicating possible strain-specific differences.

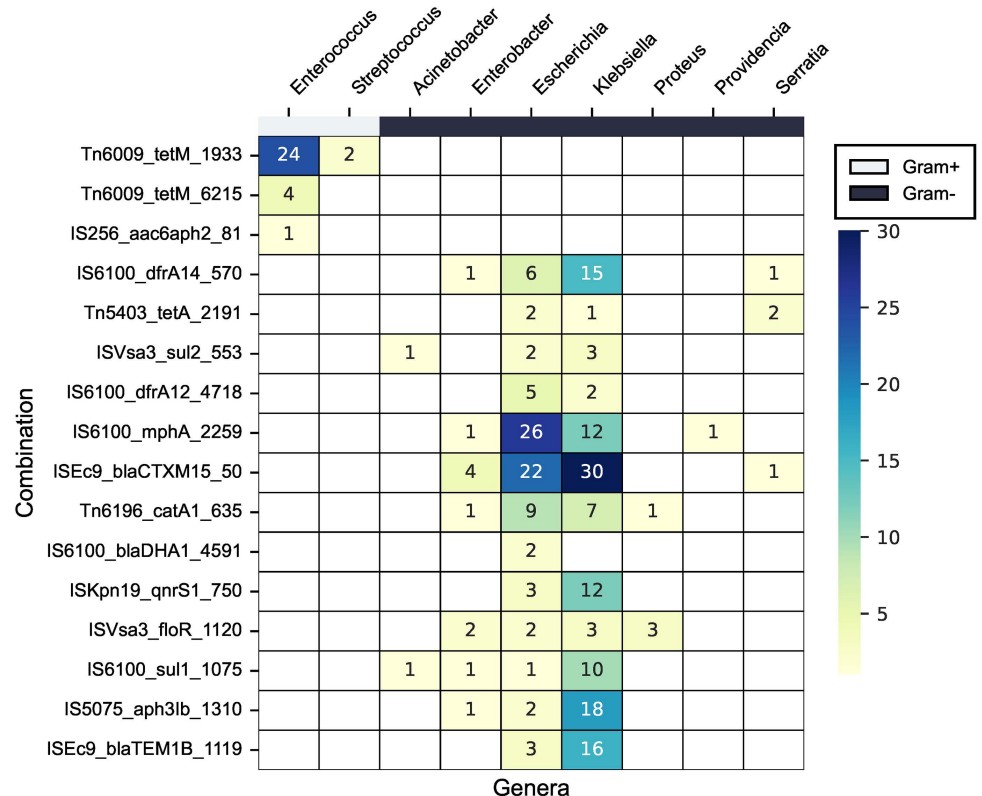

**Fig 5. Heatmap describing the number of putative translocatable (pTU) units identified in all isolates.** Rows are the pTUs; columns show the genera of the additional ~2000 isolates. The top bar display species stratified into Gram+ (white) and Gram- (black).

## iMGEs shared by multiple species

A total of 102 iMGEs were found in multiple species, with four being common between *S. aureus* and *E. faecalis* and 98 by *E. coli* and *K. pneumoniae*. Of the four iMGEs shared by Gram+ species were two IS (IS*256* and ISLmo*19*), one Tn (Tn*5405*), and one ICE (Tn*6009*), S14 Fig. IS was also the most frequent iMGE type found in *E. coli* and *K. pneumoniae* (84 elements), followed by Tns (6 elements). The Tns includes known ARG-carrying elements from the Tn*2* (Tn*2*, Tn*1000*, Tn*4656*) and Tn*3* (Tn*5403*) families. Only two ICEs (ICEKpnHS*11286* and ICEEcoED*1a*) were identified in both species.

## Differences in iMGE host ranges

The expanded isolate collection was analysed to investigate species-specific iMGE differences further. Most iMGEs were only found in related taxa, with ~39.2% of the iMGEs found in isolates of the same species, ~21.3% within the same genera, and ~19.9% within the same family (Fig 7). Interestingly, ~8.5% of the iMGEs were found in species of the same taxonomic class, and four iMGEs in multiple phyla. These included three IS (IS*6100*, ISSen*9*, and ISEfa*8*) and one Tn (Tn*6082*). The confinement of all iMGEs is presented in S4 Appendix.

The relationship between iMGEs and taxonomy was explored by creating a network that linked species together based on shared iMGEs. The network showed that related species were closely grouped in the graph as seen in S15 Fig. Four communities of densely connected species were identified with the Louvain method (Fig 8). These communities consisted of taxonomic families of the same class and Gram-staining group. For example, communities 1 and 2 consisted primarily of Gammaproteobacteria; community 1 contained Gram- and community 2 Gram+ families. However, *Listeriaceae* and

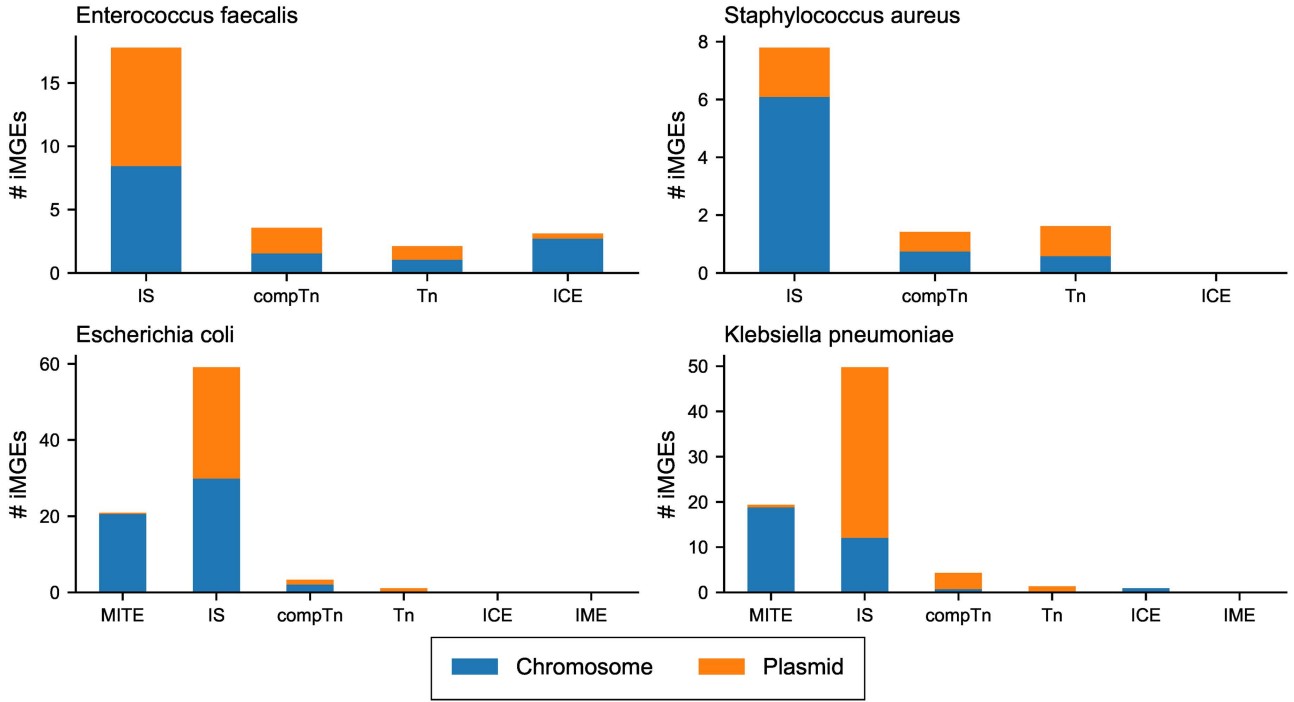

**Fig 6. The relative number of predicted iMGEs per type, genome location, and species.** The number of predicted iMGE is normalized to the number of species isolates. The bar colours show the number of MGEs predicted to be plasmid or chromosome borne. Plasmid annotation is based on annotations from plasmid prediction software.

*Corynebacteriaceae* were exceptions because they shared iMGEs with species of opposite gram-staining (ISLmo*18* by *Listeriaceae* and IS*6100*, Tn*6082* by *Corynebacteriaceae*).

## Discussion

The increasing prevalence of AMR poses a significant threat to global public health by contributing to increased mortality rates and the cost of healthcare [1,2]. MGEs play a critical role in the acquisition and dissemination of resistance genes, thereby facilitating the global spread of AMRs[6].

Here we present a comprehensive analysis of the global resistome and mobilome of four clinically significant pathogens using routine diagnostic samples collected without selection bias. This data offers an unbiased snapshot of infections occurring worldwide during 2020 [47].

Our findings revealed systematic differences in the abundance and diversity of ARGs among the four pathogens. The prevalence of certain beta-lactamase gene families and individual gene variants varied geographically, with some globally widespread while others were more prevalent in certain regions. Our findings are consistent with previous reports that the prevalence of extended-spectrum beta-lactamases (ESBL) in clinical isolates varies between different parts of the world [49]. We found the ESBL genes, such as $bla_{OXA}$ and $bla_{CTX-M}$ were more prevalent in isolates from Africa and Asia. This includes important ESBL genes such as $bla_{CTX-M-15}$ and $bla_{OXA-1}$, consistent with reports of high prevalence of ESBL in developing South Asian countries [50]. Conversely, $bla_{SHV-106}$ harboured by *K. pneumoniae,* once thought to be geographically constrained to Portugal [51], was detected across multiple continents. Our findings are supported by reports of $bla_{SHV}$-106 in Italy [52], Romania [53], and China [54], which underscores the ongoing dissemination of ESBL genes and emphasizes the need for global AMR surveillance efforts.

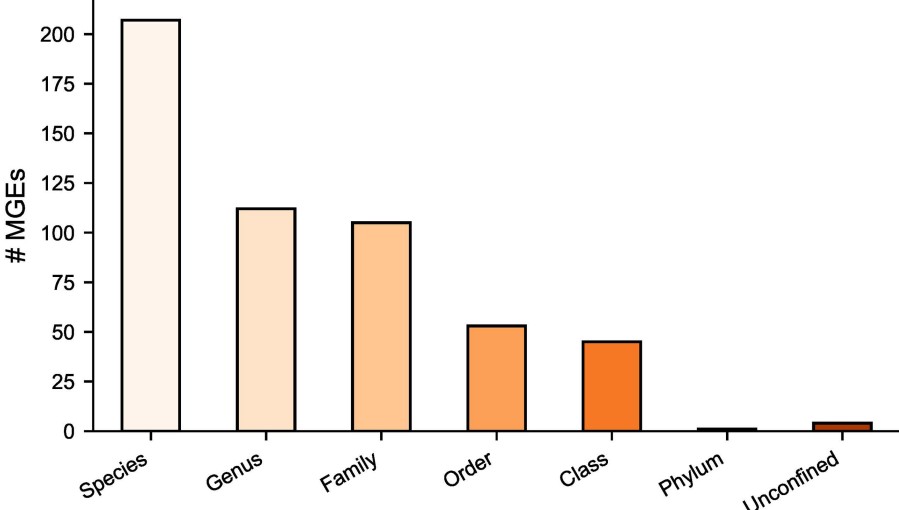

**Fig 7. Phylogenetic confinement of iMGEs.**

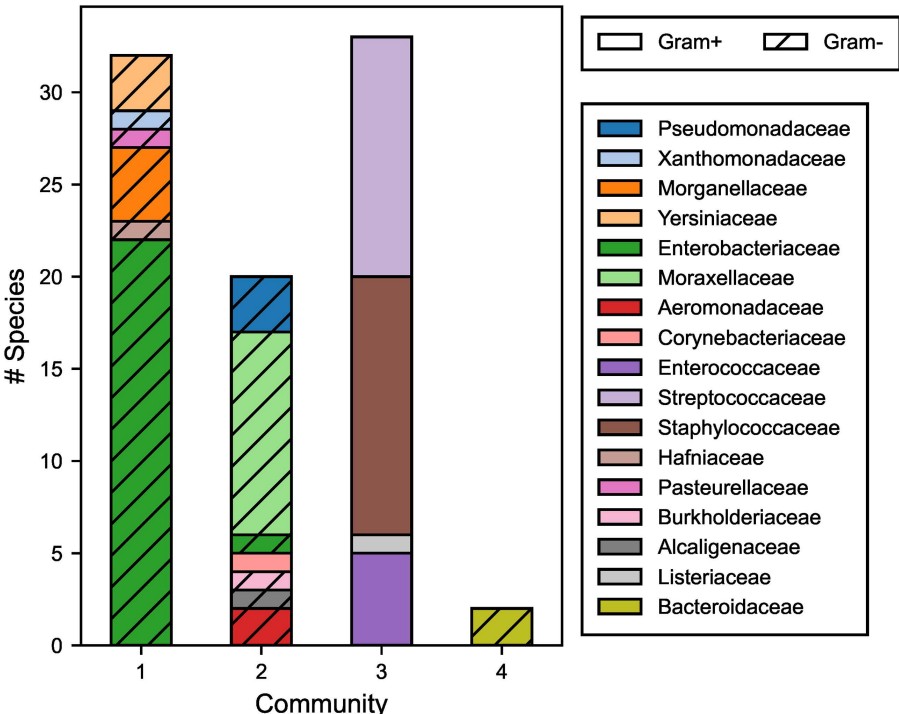

**Fig 8. Bacterial families constituting the three communities detected in the iMGE graph using the Louvain community detection method.** Communities are clusters within the graph of species densely connected by iMGEs. The bars depict the number of species coloured by their taxonomic family. Hatched bars indicate Gram- species.

We investigated the mechanisms of ARG mobility and found several genes were mobilised by plasmids and iMGEs. The number of mobilised ARGs varied depending on the bacterial species, with *E. faecalis* having the fewest. Perhaps having intrinsic resistance to common antibiotics [48,55] has resulted in *E. faecalis* being less dependent on MGEs than species with less intrinsic resistance, such as *E. coli* and *S. aureus*.

We also found evidence of 21 ARGs potentially mobilised by nearby iMGEs, with eight of these units located on chromosomal and plasmid contigs across multiple species, indicating intracellular transposition events. Many pTUs contained iMGEs previously linked to ARG mobilisation. For example, IS*6100* has been shown to mobilize resistance genes or promote genome rearrangements in *Sphingomonas* [56], *Salmonella* [57] and *Citrobacter freundii* [58]. Similarly, ISEc*9* was frequently associated with $bla_{TEM-1B}$ and $bla_{CTX-M-15}$ in pTUs found in *Escherichia*, *Klebsiella*, and *Enterobacter*, consistent with previous reports linking ISEc9 to members of the $bla_{CTX-M}$ family [59–61]. Notably, ISEc*9* has been shown to facilitate the transformation of large DNA fragments with resistance genes between clinically relevant bacteria [62] indicating a potential to facilitate horizontal gene transfer independently of conjugating elements.

We also examined the broader distribution of iMGEs across species. *E. coli* exhibited the highest diversity of both iMGEs and ARGs, possibly reflecting its large accessory genome [63]. Smaller iMGEs like MITEs and IS elements were more abundant than larger transposons and conjugative elements (ICE, IME), a pattern consistent with our previous investigation of zoonotic *Salmonella enterica* [38]. Interestingly, we observed no apparent geographic influence on MGE diversity [38], contrasting with our findings from analysing the mobilome of sewage [64].

Instead, we found that taxonomically related species tended to share similar iMGEs profiles, suggesting the distribution to be influenced by bacterial lineage and host phylogeny. This lineage-specific pattern, consistent with a previous study of zoonotic *Salmonella enterica* [38], indicates strong vertical inheritance. However, the detection of certain iMGEs across multiple phyla, including both Gram- and Gram+ species, indicates that some elements have a broad host range. These findings align with our previous studies on ARG dissimilation pathways [64–66] and suggest that such iMGEs may be particularly important in spreading resistance genes across taxonomic boundaries. Incorporating surveillance of highly transmissible MGEs into existing antimicrobial resistance frameworks may improve early detection of ARGs with high risk of being transmitted and could guide prevention strategies.

To investigate potential transmission pathways, we applied Louvain's community detection algorithm to model species connectivity based on shared iMGEs. This analysis revealed four clusters, primarily composed of bacteria of the same taxonomic class, and to a lesser extent, of the same gram-staining group. These clusters suggest gene transfer may be more likely within phylogenetically related species. While Louvain's method has proven effective in analysing biological data [67] further research is needed to determine whether such interconnections correspond with increased risk of ARG transmission.

The study was limited by our data collection and sequencing methodology. The dataset was collected in 2020 during the COVID-19 pandemic, which may have influenced the diversity and prevalence of circulating pathogens. Although the samples represented 35 countries across multiple continents, broader geographic coverage or extending the sampling period would likely provide a more comprehensive representation of the global bacterial diversity.

Technical limitations also affected our ability to characterise the MGEs fully. The use of short-read sequencing limited our ability to resolve large and complex MGEs, such as ICEs and IMEs, likely leading to their underrepresentation. To mitigate classification bias, we used the two complementary plasmid prediction tools, PlasClass and Platon, which rely on different methodologies [39,40]. Their consensus prediction aligned well with known genomic context, for example, chromosomally encoded ARGs such as *mec*A in *S. aureus* [68] and *Lsa* efflux pumps in *E. faecalis* [48] were found on chromosomal contigs. Likewise, frequently mobilized ARG families like $bla_{CTX-M}$, $bla_{TEM}$, and $bla_{OXA}$ [69–71] were predominantly located on plasmids. In addition, we expanded the MobileElementFinder database with 1,686 IS and 70 Tns to improve iMGE prediction sensitivity. Despite these limitations, the study provides valuable insight into the global resistome and mobilome of clinically relevant bacteria.

## Conclusion

MGEs can spread ARGs through multiple mechanisms, from direct carriage on plasmids to being integrated within a pTU. Our findings suggest that the transmissibility of iMGEs in pathogenic bacteria is primarily influenced by the bacterial lineage and host phylogeny. However, we identified 45 iMGEs present in species across multiple taxonomic classes, and four across multiple phyla, indicating that some elements may have a broad host range and a greater potential to disseminate ARGs across diverse bacterial populations. Additionally, we identified 21 genomic regions containing resistance genes potentially mobilised by MGEs, highlighting their role in gene transfer.

Mapping such mobilizable regions can help identify ARGs that could be transmitted. A deeper understanding of MGE transmissibility could enhance risk assessment and possibly inform future surveillance strategies. Future studies should aim to include a broader diversity of species to represent the clinical pathogens and the bacterial background better. Leveraging publicly available genomes and incorporating long-read sequencing are essential for resolving large and complex MGEs (e.g. plasmids, IMEs, and ICEs) and advancing our understanding of how these elements interact and evolve.

## Supporting information

**S1 Appendix. Sample information including accession numbers.**
(XLSX)

**S2 Appendix. ARG composition for *E. coli* and *K. pneumoniae* and their predicted genomic location.** Genes are coloured by the class of antibiotics they yield resistance to.
(XLSX)

**S3 Appendix. Name, group and familiy information of identified integrated MGEs.**
(XLSX)

**S4 Appendix. List of predicted iMGEs and in which species they were identified.**
(XLSX)

**S1 Fig. Frequency of MLST sequence types per species in the dataset.** Rare sequence types for a given species (frequency < 5%) were amalgamated into the other category. The rare sequence type constituted ~32%−80%. Map was created using shapefiles from Natural Earth.
(TIF)

**S2 Fig The number samples and location of participating diagnostic units.** a) Locations where the bacterial isolates were collected. The marker shape and colour designate the geographical region.; b) barplot showing the number of isolates per bacterial species and region.
(TIF)

**S3 Fig. Shannon diversity of iMGEs and ARGs per species.**
(TIF)

**S4 Fig. Barplot displaying the relative abundance of ARG genes aggregated by antibiotic class per species.**
(TIF)

**S5 Fig. Principal component of the ARG abundance for each species.** The scores are colored by MLST ST. The ten most common STs are displayed, and the rest are amalgamated into the other category. Ellipses shows the 90% confidence of score position.
(TIF)

**S6 Fig. Principal component of the ARG abundance for each species with scores colored by geographical region.** Ellipses show the 90% confidence of score position.
(TIF)

**S7 Fig. Boxplot showing the distribution of plasmid content for each species.**
(TIF)

**S8 Fig. Predicted ARGs and their predicted genomic location for the Gram+ species _E. faecalis_ and _S. aureus_.** Genes are coloured by the class of antibiotics they yield resistance to.
(TIF)

**S9 Fig. Principal component of the iMGE abundance coloured by species.**
(TIF)

**S10 Fig. Dendrogram showing similarity of iMGE profile of _E. coli_ isolates.** The inner circle shows MLST sequence types (ST) for the ten most frequent STs. Rare STs are amalgamated into the other category, and novel STs were omitted, showing gaps. The outer circle denotes the geographical region from which the isolate was collected. Isolates were clustered using Jaccard distance and average linkage.
(TIF)

**S11 Fig. Dendrogram showing similarity of iMGE profile of _K. pneumoniae_ isolates.** The inner circle shows MLST sequence types (ST) for the ten most frequent STs. Rare STs are amalgamated into the other category, and novel STs were omitted, showing gaps. The outer circle denotes the geographical region from which the isolate was collected. Isolates were clustered using Jaccard distance and average linkage.
(TIF)

**S12 Fig. Dendrogram showing similarity of iMGE profile of _S. aureus_ isolates.** The inner circle shows MLST sequence types (ST) for the ten most frequent STs. Rare STs are amalgamated into the other category, and novel STs were omitted, showing gaps. The outer circle denotes the geographical region from which the isolate was collected. Isolates were clustered using Jaccard distance and average linkage.
(TIF)

**S13 Fig. Dendrogram showing similarity of iMGE profile of _E. faecalis_ isolates.** The inner circle shows MLST sequence types (ST) for the ten most frequent STs. Rare STs are amalgamated into the other category, and novel STs were omitted, showing gaps. The outer circle denotes the geographical region from which the isolate was collected. Isolates were clustered using Jaccard distance and average linkage.
(TIF)

**S14 Fig. The number of iMGEs found in either Gram+ or Gram- species.**
(TIF)

**S15 Fig. Network graph showing bacterial species being connected by shared iMGEs.** Nodes represent species and are colored by family. Edges connecting nodes represent a shared iMGE.
(TIF)

**S1 Table. Number of beta-lactamase genes per continent in _E. coli_.**
(DOCX)

**S2 Table. Number of beta-lactamase genes per continent in _K. pneumoniae_.**
(DOCX)

**S3 Table. Statistics for two-way ANOVA testing difference in predicted plasmid content per species.**
(DOCX)

## Author contributions

**Conceptualization:** Markus H.K. Johansson.

**Data curation:** Markus H.K. Johansson, Sidsel Nag, Timmie M. R. Lagermann, Laura E.K. Birkedahl, Silva Tafaj, Susan Bradbury, Peter Collignon, Denise Daley, Victorien Dougnon, Kafayath Fabiyi, Boubacar Coulibaly, Réné Dembélé, Natama Magloire, Isidore J. Ouindgueta, Zenat Z. Hossain, Anowara Begoum, Deyan Donchev, Mathew Diggle, LeeAnn Turnbull, Simon Lévesque, Livia Berlinger, Kirstine K. Søgaard, Paula D. Guevara, Carolina Duarte, Panagiota Maikanti, Jana Amlerova, Pavel Drevinek, Jan Tkadlec, Milica Dilas, Achim Kaasch, Henrik T. Westh, Mohamed A. Bachtarzi, Wahiba Amhis, Carolina E.S. Salazar, José E. Villacis, Mária A.D. Lúzon, Dàmaris B. Palau, Claire Duployez, Maxime Paluche, Solomon Asante-Sefa, Mie Møller, Margaret Ip, Ivana Mareković, Agnes Pál-Sonnevend, Clementiza E. Cocuzza, Asta Dambrauskiene, Alexandre Macanze, Anelsio Cossa, Inácio Mandomando, Philip Nwajiobi-Princewill, Iruka N. Okeke, Aderemi O. Kehinde, Ini Adebiyi, Ifeoluwa Akintayo, Oluwafemi Popoola, Anthony Onipede, Anita Blomfeldt, Nora E. Nyquist, Kiri Bocker, James Ussher, Amjad Ali, Nimat Ullah, Habibullah Khan, Natalie W. Gustafson, Ikhlas Jarrar, Arif Al-Hamad, Viravarn Luvira, Wantana Paveenkittiporn, Irmak Baran, James C. L. Mwansa, Linda Sikakwa, Kaunda Yamba.

**Formal analysis:** Markus H.K. Johansson.

**Funding acquisition:** Frank M. Aarestrup.

**Investigation:** Markus H.K. Johansson.

**Methodology:** Markus H.K. Johansson.

**Project administration:** Markus H.K. Johansson.

**Resources:** Markus H.K. Johansson.

**Software:** Markus H.K. Johansson.

**Supervision:** Thomas N. Petersen, Frank M. Aarestrup.

**Validation:** Markus H.K. Johansson.

**Writing – original draft:** Markus H.K. Johansson.

**Writing – review & editing:** Markus H.K. Johansson, Thomas N. Petersen, Silva Tafaj, Frank M. Aarestrup.

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
