## [Decision Letter · Decision Letter 0]

15 Apr 2025

PONE-D-25-13754Investigation of mobile genetic elements and their association with antibiotic resistance genes in clinical pathogens worldwidePLOS ONE

Dear Dr. Johansson,

Thank you for submitting your manuscript to PLOS ONE. After careful consideration, we feel that it has merit but does not fully meet PLOS ONE’s publication criteria as it currently stands. Therefore, we invite you to submit a revised version of the manuscript that addresses the points raised during the review process.

We look forward to receiving your revised manuscript.

Kind regards,

Mabel Kamweli Aworh, DVM, MPH, PhD. FCVSN

Academic Editor

PLOS ONE

Journal Requirements:

2. Please ensure that you refer to Figure 1-8 in your text as, if accepted, production will need this reference to link the reader to the figure.

3. We notice that your supplementary figures are uploaded with the file type 'Figure'. Please amend the file type to 'Supporting Information'. Please ensure that each Supporting Information file has a legend listed in the manuscript after the references list.

**Additional Editor Comments:**

In revising your manuscript, please ensure the following key points are addressed in addition to responding to all reviewers’ comments:

**Limitations:**Please include a dedicated paragraph in the final part of your **Discussion** section clearly outlining the **limitations** of your study. This reflection should help contextualize your findings and guide interpretation.**Conclusion:**Add a **Conclusion** section at the end of the manuscript that succinctly summarizes the main findings and their implications.**Recommendations / Future Directions:**Within the Conclusion section, please include **recommendations** derived from your key findings and/or suggest potential **future research directions** that could build on your work.

Reviewers' comments:

Reviewer's Responses to Questions

**Comments to the Author**

1. Is the manuscript technically sound, and do the data support the conclusions?

Reviewer #1: Partly

Reviewer #2: Yes

Reviewer #3: Yes

2. Has the statistical analysis been performed appropriately and rigorously? 

Reviewer #1: No

Reviewer #2: Yes

Reviewer #3: Yes

3. Have the authors made all data underlying the findings in their manuscript fully available?

Reviewer #1: No

Reviewer #2: Yes

Reviewer #3: Yes

4. Is the manuscript presented in an intelligible fashion and written in standard English?

Reviewer #1: No

Reviewer #2: Yes

Reviewer #3: Yes

5. Review Comments to the Author

Reviewer #1: Comments

General comments

• The research topic is very interesting, and the subject is important. However, the presentation needs to be improved. The presentation of the methodologies could be improved. The findings should be well discussed because of the literature and not opinions.

• 74 authors are too many (though there is no cut off for the number of authors for a manuscript). However, it is clear from the author's contribution that the list from lines 734-738 are the primary authors of this research. It is suggested to the authors that all other members be acknowledged for their contribution to the paper.

Abstract

• Line 122: Are 59 diagnostic units enough to represent a snapshot of clinical infections worldwide? The methodology states 35 countries. What is the justification that 59 diagnostic units from 35 countries are enough to give a clinical representation worldwide?

• Line 142: Discussion on the identified 21 genes must be expanded

Introduction

1. Line 161: Why exclude important global pathogens like Salmonella, Campylobacter, and Listeria spp.

2. Lines requiring citations

a. Lines 165-167

b. Lines 172-174.

c. Many such sentences without appropriate citations exist in the article

3. Lines 185-187: Sentence is too long

4. Line 190: The collection of samples in a single year reflects a one-time period sampling, and this must be reflected in the abstract or title

5. Line 191: Statement is inappropriate. Symptoms of disease caused by a pathogen are more appropriate to say rather than symptoms of a pathogen

Methods

6. Line 200: Is TWIW a name or an abbreviation? Please clarify. If it is an abbreviation, the full name should be given. All such abbreviations must be defined

7. Line 211: What comprises these seven regions is undefined

8. Line 217: QC filtering may not be needed as it is deemed part of read processing

Results

9. Line 330: “(Error! Reference source not found)”. This must be corrected by supplying all appropriate citations. The same issue is found in other parts of the manuscript

10. Lines 331-337: The nomenclature of Beta-lactamase genes must be consistent following default patterns like blaCTX-M

DISCUSSION

11. Generally, this portion must be improved to really discuss the results in the context of appropriate literature providing references

12. Line 549: For abbreviation, the name should appear first, followed by the abbreviation. Example Escherichia coli (E. coli)

REFERENCES

13. The date of assessment of all Web pages and other documents without doi(s) must be provided in addition to the links

FIGURES

14. Figures all well prepared

Reviewer #2: This is an important study which aim to unravel the role of mobile genetic elements in antimicrobial resistance. I have attached some minor revisions below:

General

There are several instances of “Error reference source not found”, please review and update.

Abstract

The word count of the abstract exceeded the PLOS ONE requirement of 300 words maximum.

Introduction

The authors should include the implications of this study and how it contributes towards antimicrobial resistance solutions in public health.

Methods

Line 211: clarify what this means “and is NOT referred to as the “primary” dataset”.

Results

Line 373-375: The appropriate figure for reference is Figure S8. See the Supporting information list and caption too

Line 375-378: The appropriate figure to reference is Figure S9. See the Supporting information list and caption too.

Supporting information

The list (title and caption) for figures in the supporting information at the end of the manuscript does not match the order of their respective figures starting from Figure S3 up to S9. I will suggest a review of the figures referenced in the result section too.

Reviewer #3: Very good work and presentation however, there suggestions for improvement

1. In the Abstract, results were presented more like generalized and this was seen in the Results section of the manuscript. The authors would need to be more specific in describing that results in terms of number and/or percentages with possible denominators where necessary. Using generalized terms like "Many, Most, Some" does not give a comprehensive idea of the magnitude of ARGs, iMGEs including their diversity in the species of interest. I could see that the charts and table show some quantitative results but it will be good to include them in the results description as well as put them in context.

2. The missing references sources should to fixed by the authors. It makes the article easier to follow should the review wants to check the cited/referenced articles.

3. The section on "Controlling for high relatedness in the dataset" can be moved to methods because it is not actually describing the results. It describes more of methods. Same as the opening paragraph in the results except the authors consider rephrasing them to actually mean the results if they must stay in the result section.

6. PLOS authors have the option to publish the peer review history of their article (what does this mean? ). If published, this will include your full peer review and any attached files.

**Do you want your identity to be public for this peer review?** For information about this choice, including consent withdrawal, please see our Privacy Policy .

Reviewer #1: No

Reviewer #2: **Yes: ** Damilola Odumade

Reviewer #3: No

---

## [Author Response · Author response to Decision Letter 1]

15 Jun 2025

Editor comments

Response to comment 1:

We added a dedicated paragraph to discuss the limitations of the study.

Response to comment 2:

We added a conclusion section at the end of the manuscript. The conclusion outlines directions for future research.

Response to comment 3:

See the response above.

Reviewer 1

=======

Response to the general comments:

We share the reviewer’s concern regarding the large number of authors. However, we also have to realise that collection and organising local sampling and providing the necessary epidemiological and context information is a large task. It was before initiating the study agreed with all partners that we all would be co-authors on the manuscript.

Regarding the comment on dataset representation. It is true that we have not collected data from all world countries but only a representative set of 35 countries. We chose to describe data as a worldwide snapshot of clinical infections due to sampling from 6 world regions. We are discussing the limited number of countries in the updated discussion and have included number of countries in the abstract.

Additionally, we revised and expanded the discussion on pTUs and their associated iMGEs.

Response to comment 1:

Our intention was not to exclude important pathogens, but instead to collect a “snapshot” of bacterial pathogens encountered by the participating diagnostic units. Because the samples were not selected based on species or strains, important pathogens might only be represented by a few samples in the dataset. For example, the dataset includes one Listeria monocytogenes sample and 36 Salmonella enterica samples, of which 21 were from the same institute in Colombia.

Response to comment 2:

We added additional citations when needed.

Response to comment 3:

Broke up the sentence. It now reads “The interplay of different MGEs forms a complex transposition network that has been essential for recruiting ARGs[10,20] and spreading them to infectious bacteria[21,22]. MGEs are also thought to help retain resistance genes in environments with low levels of antimicrobials.”

Response to comment 4:

We have added the collection year to the manuscript.

Response to comment 5:

Changed the statement to say “symptoms of a disease caused by a pathogen”

Response to comment 6:

Wrote out the full name of the collaboration.

Response to comment 7:

Changed to “the 6 WHO defined regions as previously described[23]”

Response to comment 8:

Deleted “passed QC”

Response to comment 9:

We reviewed and corrected broken cross-references in the manuscript.

Response to comment 10:

Corrected the formatting of beta-lactamase genes and families to follow the nomenclature.

Response to comment 11:

We agree with the reviewer and have reworked the discussion to increase clarity and to put our findings in context of relevant research.

Response to comment 12:

The abbreviations were reviewed and corrected when needed.

However, abbreviated species names were not written the first time a species name was used as it would not to conform to PLOS style guides.

Response to comment 13:

Dates when webpages and other online resources were accessed have been added.

Reviewer 2

=======

Response to comment 1:

The broken references have been updated.

Response to comment 2:

We shortened the abstract to 298 words.

Response to comment 3:

We agree with the reviewer and have revised the discussion to better put our findings into the context of public health.

Response to comment 4:

Corrected to “now referred to as…”

Response to comment 5:

We reviewed and updated references and captions.

Response to comment 6:

See response to comment 5.

Response to comment 7:

See response to comment 5.

Reviewer 3

=======

Response to comment 1:

We reworked the abstract to be more specific and comply with the word limit.

Response to comment 2:

We reviewed and fixed the broken references.

Response to comment 3:

We agree with the reviewer’s suggestion and have moved the section to Materials and Methods and reworked the opening paragraph of the Results for increased clarity.

---

## [Decision Letter · Decision Letter 1]

1 Jul 2025

PONE-D-25-13754R1Investigation of mobile genetic elements and their association with antibiotic resistance genes in clinical pathogens worldwidePLOS ONE

Dear Dr. Johansson,

Thank you for submitting your manuscript to PLOS ONE. After careful consideration, we feel that it has merit but does not fully meet PLOS ONE’s publication criteria as it currently stands. Therefore, we invite you to submit a revised version of the manuscript that addresses the points raised during the review process.

We look forward to receiving your revised manuscript.

Kind regards,

Mabel Kamweli Aworh, DVM, MPH, PhD. FCVSN

Academic Editor

PLOS ONE

**Journal Requirements:**

Reviewers' comments:

Reviewer's Responses to Questions

**Comments to the Author**

1. If the authors have adequately addressed your comments raised in a previous round of review and you feel that this manuscript is now acceptable for publication, you may indicate that here to bypass the “Comments to the Author” section, enter your conflict of interest statement in the “Confidential to Editor” section, and submit your "Accept" recommendation.

Reviewer #1: All comments have been addressed

Reviewer #2: All comments have been addressed

Reviewer #3: All comments have been addressed

2. Is the manuscript technically sound, and do the data support the conclusions?

Reviewer #1: Yes

Reviewer #2: Yes

Reviewer #3: Yes

3. Has the statistical analysis been performed appropriately and rigorously? 

Reviewer #1: Yes

Reviewer #2: N/A

Reviewer #3: N/A

4. Have the authors made all data underlying the findings in their manuscript fully available?

Reviewer #1: Yes

Reviewer #2: Yes

Reviewer #3: Yes

5. Is the manuscript presented in an intelligible fashion and written in standard English?

Reviewer #1: Yes

Reviewer #2: Yes

Reviewer #3: Yes

6. Review Comments to the Author

**Reviewer #1: ** All comments have been addressed by the authors and I recommend accepting the manuscript for publication

**Reviewer #2: ** Thank you for addressing the previous comments. However, there are a few additional issues that need to be resolved:

Figure 1 – Geographic distribution of β-lactamase gene families in E. coli

The current map in Figure 1 does not include the African continent, which creates an inconsistency (compare with Figure 2) and Africa is explicitly referenced in Line 309-310. I recommend updating the figure to include Africa in order to accurately reflect the geographic scope discussed in the text.

Line 354 – Inconsistent Figure Referencing

The current reference style in this line — “S8 Fig S8” — appears inconsistent. I suggest the authors adopt a consistent and standard format for referencing all figures throughout the manuscript (e.g., “S8 Fig” or “Fig S8,” not both), and apply this uniformly across the document.

**Reviewer #3: ** The authors have made all corrections that I recommended that the editors review for guidelines and then publish.

7. PLOS authors have the option to publish the peer review history of their article (what does this mean? ). If published, this will include your full peer review and any attached files.

**Do you want your identity to be public for this peer review?** For information about this choice, including consent withdrawal, please see our Privacy Policy .

Reviewer #1: No

Reviewer #2: **Yes: ** Damilola Odumade

Reviewer #3: No

---

## [Author Response · Author response to Decision Letter 2]

27 Jul 2025

Thank you for the opportunity to revise our manuscript “Investigation of mobile genetic elements and their association with antibiotic resistance genes in clinical pathogens worldwide” (Submission ID: PONE-D-25-13754). We are grateful for the thoughtful and constructive feedback, which has helped us to improve the clarity and accuracy of our work.

We sincerely thank the reviewer for their valuable feedback and hope that the revised version meets the expectations of the reviewers and editorial team.

Sincerely,

Markus Johansson

Reviewer 2

Comment 1

Figure 1 – Geographic distribution of β-lactamase gene families in E. coli

The current map in Figure 1 does not include the African continent, which creates an inconsistency (compare with Figure 2) and Africa is explicitly referenced in Line 309-310. I recommend updating the figure to include Africa in order to accurately reflect the geographic scope discussed in the text.

Response:

We agree with the reviewer's observation. We have redrawn Figure 1 to include the African continent, ensuring that it reflects the data being discussed in the text and that the figure is consistent with Figure 2.

Comment 2

Line 354 – Inconsistent Figure Referencing

The current reference style in this line — “S8 Fig S8” — appears inconsistent. I suggest the authors adopt a consistent and standard format for referencing all figures throughout the manuscript (e.g., “S8 Fig” or “Fig S8,” not both), and apply this uniformly across the document.

Response:

Thank you for pointing this out. We have carefully reviewed the manuscript and corrected all inconsistent figure references to ensure uniformity. The following changes were made:

Line 303; “S3 Fig S3b” was changed to “S3b Fig”

Line 334; “Fig S4” was changed to “S4 Fig”

Line 340; “Fig S5” was changed to “S5 Fig”

Line 341; “Fig S6” was changed to “S6 Fig”

Line 348; “Fig S7” was changed to “Fig S7”

Line 354; “S8 Fig S8” was changed to “S8 Fig”

Line 356; “Fig S9” was changed to “S9 Fig”

Line 398; “Fig S3” was changed to “S3 Fig”

---

## [Editor Report · Decision Letter 2]

30 Jul 2025

Investigation of mobile genetic elements and their association with antibiotic resistance genes in clinical pathogens worldwide

PONE-D-25-13754R2

Dear Dr. Johansson,

We’re pleased to inform you that your manuscript has been judged scientifically suitable for publication and will be formally accepted for publication once it meets all outstanding technical requirements.

Kind regards,

Mabel Kamweli Aworh, DVM, MPH, PhD. FCVSN

Academic Editor

PLOS ONE
---

## [Editor Report · Acceptance letter]

PONE-D-25-13754R2

PLOS ONE

Dear Dr. Johansson,

I'm pleased to inform you that your manuscript has been deemed suitable for publication in PLOS ONE. Congratulations! Your manuscript is now being handed over to our production team.

Kind regards,

on behalf of

Dr. Mabel Kamweli Aworh

Academic Editor

PLOS ONE